# T-cell activation and senescence in asymptomatic HIV/*Leishmania infantum* co-infection

**Carolina de Oliveira Mendes-Aguiar[1☉], Manoella do Monte Alves[2☉],
Amanda de Albuquerque Lopes Machado[1], Glória Regina de Góis Monteiro[1],
Iara Marques Medeiros[1], Jose Wilton Queiroz[1†], Iraci Duarte Lima[1], Eliardo G. Costa[3],
Richard D. Pearson[4], Mary E. Wilson[5], Marshall J. Glesby[6],
Eliana Lúcia Tomaz do Nascimento[1,2], Selma Maria Bezerra Jerônimo[1,7,8]***

**1** Institute of Tropical Medicine of Rio Grande do Norte, Federal University of Rio Grande do Norte, Natal, Brazil, **2** Department of Infectious Disease, Health Science Center, Federal University of Rio Grande do Norte, Natal, Brazil, **3** Department of Statistics, Federal University of Rio Grande do Norte, Natal, Rio Grande do Norte, Brazil, **4** Division of Infectious Diseases and International Health, Department of Medicine, University of Virginia, Charlottesville, Virginia, United States of America, **5** Departments of Internal Medicine and Microbiology & Immunology, University of Iowa and the Veterans' Affairs Medical Center, Iowa City, Iowa, United States of America, **6** Division of Infectious Disease, Weill Cornell Medical College, New York, New York, United States of America, **7** Institute of Science and Technology of Tropical Diseases, Natal, Brazil, **8** Department of Biochemistry, Federal University of Rio Grande do Norte, Natal, Brazil

† Deceased.
☉ These authors contributed equally to this work.
* selma.jeronimo@ufrn.br

## Abstract

### Background

*Leishmania infantum* can be an opportunistic pathogen, with an immunocompromised status increasing the risk of converting asymptomatic infection to symptomatic visceral leishmaniasis (VL). VL has approximately 5% fatality rate; and HIV coinfection (AIDS/VL) increases this risk. We hypothesized that, relative to those with HIV alone, people with co-infection would have altered T cell activation which could impact on the risk of VL.

### Methods

A cross-sectional study was performed between 2014 and 2016 to determine the prevalence of *L. infantum* infection in people living with HIV (PLHIV) residing in Brazil (n = 1,372). Subsequent incident cases of VL were ascertained from a public health database through 2018 and from a cohort of families with VL. Immune status of 69 participants was evaluated and comparisons made between those with and without HIV, with latent or with active Leishmania infection and those without HIV but with active or resolved Leishmania or T cell hypersensitivity to Leishmania antigen and healthy control subjects.

**Data availability statement:** All data are in the manuscript and/or Supporting Information files.

**Funding:** The work was supported from a grant from the CNPq (universal 406076/2021-9) awarded to SMBJ and National Institute of Science and Technology of Tropical Diseases (INCT-DT, 465229/2014-0) awarded to EMC and SMBJ. The funders had no role in study design, data collection and analysis, decision to publish, or preparation of the manuscript.

**Competing interests:** The authors have declared that no competing interests exist

## Results

A total of 24.2% of PLHIV had positive anti-IgG *L. infantum* antibodies. The relative risk of developing AIDS/VL was 2.27 (95% CI: 0.920 to 5.59; p = 0.07) to HIV/Leish coinfected subjects with positive leishmania serology compared to HIV subjects without leishmania serology. Poor adherence to antiretroviral therapy (p = 0.0008) or prior opportunistic infections (p = 0.0007) was associated with development of AIDS/VL in asymptomatic HIV/Leish. CD4$^+$ and CD8$^+$ T cells counts or viral load were similar between asymptomatic HIV/Leish and HIV subjects. However, activated CD8$^+$CD38$^+$HLA-DR$^+$ T cells were higher in asymptomatic HIV/Leish than HIV. Likewise, senescent (CD57$^+$) and PD1$^+$ CD8$^+$ T cells were higher in asymptomatic HIV/Leish than in AIDS/VL or HIV groups.

## Conclusion

Although asymptomatic HIV/Leish subjects had CD4$^+$ and CD8$^+$ T cells similar to HIV alone, their CD8$^+$T cells had increased activation and senescence which could contribute to risk of developing VL.

### Author summary

The frequency of asymptomatic HIV/*Leishmania infantum* (HIV/Leish) infections and the immunological status of subjects with HIV residing in the state of Rio Grande do Norte, Brazil, between 2014 and 2016 were studied. A high frequency of asymptomatic HIV/*Leishmania infantum* infections (PLHIV with positive anti-IgG *L. infantum* antibodies) was found. Asymptomatic HIV/Leish subjects had CD8$^+$ T cells with higher markers of activation and senescence than PLHIV with negative serology to *L. infantum,* however, they were similar to AIDS/VL. At the time of this study, there was still poor availability of antiretroviral therapy. Asymptomatic HIV/Leish had high relative risk of developing AIDS/VL. Thus, PLHIV residing in endemic areas for VL should be assessed for their *L. infantum* infection status and advised to closely adhere to antiretroviral therapy to decrease the risk of developing VL or any other co-morbidity.

## Introduction

Visceral leishmaniasis (VL) is a prevalent disease in many tropical and subtropical areas of the world, with more than 90% of cases reported from India, Nepal, Bangladesh, Sudan and Brazil. *Leishmania infantum* is the main etiologic agent of VL in Latin America and Europe [1]. *L. infantum* infection outcomes vary from asymptomatic to severe disease, but asymptomatic infection occurs in most people. In endemic regions of Brazil, an improvement in nutrition and vaccination rate coverage for children and better sanitary conditions resulted in an increased trend to asymptomatic infection rather than symptomatic VL; in addition, there has been a decrease of disease among children [2–4]. The factors determining the conversion from asymptomatic infection to VL are still not entirely understood.

Studies indicate that high antibody titers are more often seen in subjects with symptomatic VL [2,3]. Subsets of T cells from asymptomatic *L. infantum* infected subjects are capable of activation after *L. infantum* antigens stimulation, detected by the marker CD69$^+$. The majority of asymptomatic subjects eventually mount a protective Th1-type cellular immune responses

with IL-2 and IFN-γ production, that is long-lasting if there is no immunosuppression, possibly indicating protective immunity [4]. In contrast, symptomatic VL subjects have unique T cell responses characterized by exhaustion and activation profiles, with decreased IFN-γ, TNF and IL-6; and increase IL-10 production [5]. A cytokine storm was observed in VL patients, similar to observed in sepsis, with elevated levels of anti-and pro-inflammatory cytokines, and this phenomenon was associated with elevated levels of LPS, sCD14 and I-FABP, and indicative of microbial translocation [6]. A similar immune profile is seen in dogs with VL and usually these animals do not have self-resolving infection and have increased risk of death [7].

The resurgence of VL in Europe occurred concurrently with the HIV epidemic, first spread through sharing of needles, but also through the expansion of canine VL in several countries including Portugal, Spain, France and Italy. In Brazil, HIV was initially an infection of urban settings and VL was a disease predominantly in rural areas. However, in the last 30 years VL has become frequent in peri-metropolitan areas of major cities in the Northeast and Southeast regions of the country, and HIV has spread to rural areas. The increased regional overlap of both infections has led to changes in the epidemiology of VL due to co-infection [1,8]. In the state of Rio Grande do Norte, Brazil, this resulted in an increase in AIDS/VL cases detected after 2010 [8,9].

A high risk of VL development was observed among people living with HIV (PLHIV) [10], and *L. infantum* is considered an opportunistic pathogen in this population [11]. The impact of VL in people who are immunocompromised by HIV can be devastating, since both diseases suppress the immune system [12]. Asymptomatic co-infection with HIV and *L. infantum* increases the risk of VL development [13], and in Brazil co-infected patients have had more VL relapses and higher rates of death than non-HIV VL [1]. Relapses of VL have been associated with falling CD4+ T cell counts [10].

*Leishmania* infection increases HIV viral load in serum from co-infected people [14]. Consistently, macrophages co-infected with HIV *in vitro* have higher *L. infantum* loads and increased capacity to uptake the mammalian stage, the amastigote [15,16]. Chronic HIV infection is known to lead to CD4+ T cell depletion and CD8+ T cell activation [12]. Cellular immune responses are essential for control of *Leishmania* spp. infection, with contributions of CD4+ T cells to the outcome of active disease and CD8+ cells contributing to immune protection. It is possible that the low CD4+ T cells characterizing HIV infection synergize with the suppressed hematopoietic cell functions due to *Leishmania* spp. infection [17]. Furthermore, AIDS/VL co-infected subjects exhibited increased CD8+ T cell activation during active VL and after remission compared to people with HIV alone [18]; the chronic T cell activation phenotype was positively correlated with T cell senescence, leading to accelerated immune cell impairment [17].

Our overreaching hypothesis was that asymptomatic individuals co-infected HIV/Leish would be treated as HIV without positive Leishmania serology individuals, however, would present an altered immunological profile, similar to individuals with AIDS/VL and, therefore, would have a higher risk of developing AIDS/VL. The immunologic profile of PLHIV with positive serology for *L. infantum* has not previously been described. We analyzed CD4+ and CD8 +T cell activation and senescence in HIV, asymptomatic HIV-*L. infantum* co-infection, AIDS/VL co-infection, as well as, immunocompetent groups, active symptomatic VL, Recovered VL, DTH+ and healthy subjects. Our results described for the first time the immunological profile of PLHIV with positive serology. Moreover, we observed an increased risk of development of AIDS/VL by asymptomatic HIV/*L. infantum* co-infected subjects (PLHIV with latent *L. infantum* infection) compared to HIV (PLHIV with negative *L. infantum* serology) after three years since the first examination. Of importance, this HIV population was studied prior to the era to the initiation of ART in Brazil. The presence of HIV infection

was associated with PD1+CD8+T cells, and HIV viral load was negatively correlated with CD4+ counts. Moreover, *L. infantum* infection influenced activated CD4+T, PD1+CD8+T and senescent CD8+T cells. Suboptimal use of ART and previous history of opportunistic infection were associated with an increased risk of AIDS/VL outcome. Thus, PLHIV residing in areas endemic for visceral leishmaniasis area should have measurement of *L. infantum* serology and close follow-up with an emphasis on the importance of consistent use of ART.

## Methods

### Ethical considerations

The protocol and informed consent were reviewed and approved by the Federal University of Rio Grande do Norte Ethical Committee (CEP-UFRN) and by the Brazilian National Ethics Committee (CONEP). The certificate of ethical approval is CAAE 12675013.7.0000.5537. Informed consent was obtained, and the consent form was signed by the participant or their legal guardian before any study-related procedures. Exemption of consent was obtained for those from the cross-sectional study when contact was not possible and were not included in the repeated blood collections

### Study design and population

1. A cross-sectional study of PLHIV to assess co-infection with *Leishmania infantum*. The study was performed between July 2014 and January 2016, in which 1,372 people living with HIV residing in Rio Grande do Norte State, Brazil, were tested for *L. infantum* infection by SLA (soluble leishmania antigens) prepared from a local isolate and rK39 antigens using ELISA [18]. Of importance, at the time of recruitment, in 2014, antiretroviral therapy was recommended for those with CD4+ cell counts below 350 cells/mm³ [19].

2. Retrospective analysis of AIDS-VL. A retrospective analysis was performed in 2019 considering the data on symptomatic VL cases which were obtained from the Notifiable Diseases Information System (SINAN) covering the period from July 2014 to December 2018 in order to ascertain the diagnosed AIDS/VL cases for the study period. AIDS/VL reported cases in SINAN database were cross referenced to 1,372 HIV cases evaluated in the cross-sectional study of PLHIV. The hypothesis herein was that since the HIV population recruited between 2014-2016 had not started promptly on antiretroviral therapy, they were more likely to develop AIDS-VL, once they became infected with *L. infantum*.

3. Immunological studies to assess T cell function. A total of 69 subjects were recruited and their group allocations are shown in Table 1. HIV (n = 16) and Asympt HIV/Leish (n = 10) subjects were drawn from the cross-sectional HIV study, and these subjects had a second blood sample collected for further serological evaluation and for immunophenotyping of T cells for exhaustion, senescence and activation. People with AIDS/VL and symptomatic VL were recruited at Giselda Trigueiro Hospital, Natal, Brazil, during their treatment for VL; whereas Recovered VL and DTH subjects were recruited from a cohort study of families with a history of a symptomatic case. This latter population has been described previously [20,21]. Uninfected Controls were in the same endemic area from which VL and recovered VL were recruited.

### ELISA serologic test for asymptomatic *L. infantum* case detection

People living with HIV, Asympt HIV/Leish, AIDS/VL, VL, Recovered VL, DTH+ and UC subjects were screened for the presence of anti-Leishmania IgG antibodies using two sources of *L. infantum* antigens. (1) A soluble lysate of *L. infantum* (SLA) was produced from an

**Table 1. Subject groups undergoing immunologic testing.**

| Group | Group name | n | Definition |
|---|---|---|---|
| HIV only | **HIV** | 16 | PLHIV on ART, with negative *L. infantum* serology (SLA and/or rK39) and no symptoms of VL |
| Asymptomatic HIV+/*L. infantum+* | **Asympt HIV/Leish** | 10 | PLHIV on ART, positive *L. infantum* serology (SLA and/or rK39) and no symptoms of VL |
| AIDS/VL | **AIDS/VL** | 14 | AIDS and symptomatic VL* |
| Symptomatic VL | **VL** | 8 | Symptomatic VL* and HIV negative |
| Recovered VL | **Recovered VL** | 6 | Immunocompetent without HIV subjects treated for VL who had clinical recovery |
| DTH+ | **DTH+** | 9 | Immunocompetent without HIV subjects with DTH skin test response to *L. amazonensis* antigens (or leishmanin skin test) with no history of VL |
| Healthy Control/ Uninfected Controls | **UC** | 6 | Healthy controls with negative serology to *L. infantum* (SLA and/or rK39) and negative DTH skin test response to *L. amazonensis* antigens with no history of VL or HIV infection |

*The VL status was confirmed by either parasitological findings in the bone marrow aspirate or by presence of anti- *L. infantum* antibodies and with clinical findings consistent with the disease.

isolate from a patient with VL in Natal whose Leishmania isolate was typed in a World Health Organization Reference Laboratory (Elisa Cupolillo, PhD, Fiocruz, Rio de Janeiro, Brazil). (2) rK39 antigen for ELISA was kindly provided by Steven G. Reed (Infectious Diseases Research Institute, Seattle, WA). Both ELISA protocols have been previously described [18,22]. The cutoff for a positive reaction was determined as the mean + three standard deviations of the absorbance of negative control sera (minus absorbance in blank wells). Each serum sample was analyzed in duplicate. The results are shown as relative OD (rOD), normalized to the cut-off for each. The rOD was determined as the (mean of sample duplicates) – (mean absorbance in blank wells)/ Cut-off. The results were expressed by rOD median/min-max; p value.

## Delayed type hypersensitivity skin test (DTH) to *Leishmania* antigens

The antigen used for assessment of the DTH skin test was kindly provided by the Centro de Produção e Pesquisa de Imunobiológico/PR (CPPI, Paraná, Brazil). It was prepared from *L. amazonensis*, and it was the only DTH preparation approved for clinical use in Brazil. Of importance, the only circulating Leishmania in the area is *L. infantum*, and previous study performed using this preparation showed high sensitivity to identify previous *L. infantum* infection [5,23]. The antigens were placed and read in accordance with the ballpoint pen technique [24]. Induration greater than 5 mm in diameter was considered positive [25].

## Lymphocyte isolation and flow cytometric detection of T cell subsets

Two milliliters of peripheral blood were collected and red blood lysis buffer was added. After red blood cell lysis, white cells were washed by centrifugation in 0.1% PBS-Azide and then labeled with monoclonal antibodies. Activation status (CD38-FITC; BD, clone HIT2/ HLA-DR-PE; Ebioscience, clone L243), Senescence (CD57-FITC; BD, clone NK-1) and PD-1-PERCP (Biolegend, clone NAT105) were analyzed in $CD4^+CD3^+$ and in $CD8^+CD3^+$ T cells by flow cytometry. After antibody incubation, cells were fixed in 1% formol buffer. Thirty thousand events were acquired from each sample using a FACS Canto II flow cytometer (BD, Canto II flow cytometer, Becton Dickinson Bioscience, USA). FlowJo vX software was used for flow cytometry analysis. Lymphocytes were gated in forward scatter and side scatter dot plot graphic. Activation, senescence and exhaustion markers were evaluated in

gated CD4$^+$CD3$^+$ and in CD8$^+$CD3$^+$ T lymphocyte subpopulation. Results were expressed as percentage (median; min-max).

## Statistical analysis

Continuous variables were expressed as medians and interquartile ranges. Kruskal-Wallis nonparametric test with multiple comparisons was used to analyze differences between groups using GraphPad Prism software (version 8.0, San Diego, CA, USA). Chi-square test was performed to compare categorical variables. Differences were considered significant when the p value was <0.05. We evaluated strengths of relationships between two continuous variables using Spearman's correlation coefficient (rho). Relative risk of developing AIDS/VL was computed with 95% confidence intervals (Cl) [26]. To compare the phenotypes, as described in Table 1, regression models were used. When the response variable was a positive real number, we assumed an inverse gaussian or a gamma distribution with an identity link (the mean equals to the regression equation) and when the response variable was non-negative integer values, Poisson or a negative binomial distribution was assumed [27,28]. When the support of the response variable was a proportion, we fit a beta regression model with a log link [29] or we treated the response variable as a percentage and fit a generalized linear model assuming a gamma distribution with identity link. To assess the assumptions of the regression models we computed the standardized quantile residuals for each fitted model, and we obtained plots of the residuals against fitted values and index, autocorrelation plots of the residuals and normal quantile-quantile plots with simulated envelope of the residuals. We computed Cook's distance in order to detect influential observations [30]. Pairwise comparisons of the estimated marginal means obtained from the final fitted regression models were performed. Finally, principal component analysis (PCA) of all immunological parameters was done using GraphPad Prism 9.0. SLA and rK39 rOD and frequency of activated, senescent and PD1+ TCD4+ and TCD8+ cells were selected as variables, and Kaiser rule was used to evaluate PC1, PC2 and PC3. PC scores were used to evaluate presence of clusters in groups variables using K-means method and silhouette plot to assess the existence of clustering. We also used the R language [31] to do the computations. For the final fitted regression models, the assumptions of the model were reasonably valid and no influential observations were detected.

## Results

### *Leishmania infantum* infection in PLHIV in endemic area of Rio Grande do Norte, Brazil, and visceral leishmaniasis development

PLHIV (n = 1,372) were tested for *L. infantum* infection, using SLA and rK39 anti-IgG ELISA serology. Of those, 333 (24.2%, 333/1,372) had positive antibodies to rK39 and/or SLA. Anti-Leishmania antibodies using SLA or rK39 were detected in 17.2% (236/1,372) or 12.9% (178/1,372) subjects, respectively. SLA rOD results correlated with rK39 rDO (r = 0.82; p<0.0001).

Three years after the initial Leishmania serology screening, 19 (1.38%) subjects developed visceral leishmaniasis (AIDS/VL) and were treated for VL. Among the subjects who developed AIDS/VL, 11 had initially tested negative for *L. infantum* infection (11 of 1,039; 1.05%) and 8 had tested positive for *L. infantum* (8 of 333; 2.4%) (Chi-square; p = 0.033). The relative risk of developing AIDS/VL for those participants living with HIV and with positive Leishmania serology versus these without positive Leishmania serology after three years from the beginning of the study was 2.27 (CI95%: 0.920 to 5.59; p = 0.07).

### Immunologic and virologic assessments comparing HIV and HIV-asymptomatic *L. infantum* infected group

The immunologic and virologic status of a subgroup of asymptomatic HIV/Leish subjects was compared to HIV and AIDS/VL subjects. Results are shown in Table 2, as median (min-max). Four of the seven groups had a male predominance while the remainder were more evenly balanced. Most subjects in each group were male (VL (87.5%), AIDS/VL (85%) and asymptomatic HIV/Leish individuals (70%)). There was no difference in the age between groups (Kruskal-Wallis p = 0.56) (Table 2).

All Asympt HIV/Leish individuals (10 of 10) had positive anti-IgG SLA serology, and 5 of 10 individual had positive anti-IgG rK39 serology. Asympt HIV/Leish individuals had higher anti-SLA IgG antibodies and similar levels of anti-rK39 IgG antibodies compared to AIDS/VL (Tables 2 and S1). VL patients had higher levels of anti- *L. infantum* antibodies than AIDS/VL group (SLA and rK39) or DTH+ subjects (SLA). Most AIDS/VL subjects had negative serology to *L. infantum* antigens (SLA: 71.4%/ rK39: 78.5%). DTH+ subjects had similar rOD antibodies levels as Asympt HIV/Leish individuals or recovered VL subjects. SLA rOD correlated with rK39 rDO (r = 0.88; p<0.0001). HIV groups differed with respect to their HIV viral loads and CD4$^+$ T cell counts, but not their CD8$^+$ T cell counts (Tables 2 and S2). HIV and Asympt HIV/Leish individuals had lower viral loads than individuals with AIDS/VL. At the same time, CD4$^+$ T counts were higher in HIV and in Asympt HIV/Leish individuals than in AIDS/VL. Viral load, and CD4$^+$ and CD8$^+$ T cell counts did not differ between HIV and Asympt HIV/Leish individuals. HIV viral load was negatively correlated with CD4$^+$ count (r = 0.53; p = 0.005), while CD4$^+$ count was positively correlated with CD8$^+$ count (r = 0.45; p = 0.01). No differences in CD8$^+$ T cell counts were observed between the HIV, Asympt HIV/Leish and AIDS/VL groups (ANOVA Kruskal-Wallis p = 0.25) (S2 Table).

ART was used by 81.25% (13/16) of the HIV subjects. However, in *L. infantum* infected groups, the adherence to ART was lower. Sixty percent (6/10) of asympt HIV/Leish subjects and 7.15% (1/14) of AIDS/VL subjects used ART regularly. Subjects were asked about their history of prior infections with *Toxoplasma* spp., *Mycobacterium tuberculosis* or *Mycobacterium leprae*. Before enrollment, 25% (4/16) of HIV subjects had had one of these infections

**Table 2. Clinical and laboratory characteristics of patients included in the immunologic study.**

| Groups | Age (Years) | Gender (%male) | CD4+T cells count (cells/mm³) | CD8+T cells count (cells/mm³) | HIV Viral load (Log$_{10}$copies/mL) | rOD anti-IgG SLA | rOD anti-IgG rK39 |
|---|---|---|---|---|---|---|---|
| UC (n = 6) | 45 (35–52) | 3F/3M (50.0%) | ND | ND | ND | 0.15 (0.13–0.7) | 0.29 (0.15–0.7) |
| HIV (n = 16) | 42 (21–54) | 9F/7M (43.7%) | 620 (64–1,153) | 817 (284–1,992) | UND (UND–3.9) | 0.17 (0.14–0.49) | 0.33 (0.16–0.61) |
| Asympt HIV/Leish (n = 10) | 41 (29–62) | 3F/7M (70.0%) | 414 (105–747) | 1002 (316–3,992) | UND (UND–4.5) | 1.66 (1.01–3.86) | 0.96 (0.35–2.97) |
| AIDS/VL (n = 14) | 40 (18–53) | 2F/12M (85.0%) | 85 (8–161) | 737 (218–1,194) | 5.3 (UND–7.0) | 0.55 (0.13–6.17) | 0.59 (0.19–7.91) |
| VL (n = 8) | 35 (16–51) | 1F/7M (87.5%) | ND | ND | ND | 3.93 (1.63–10.47) | 5.79 (1.02–9.47) |
| Recovered VL (n = 9) | 28 (15–32) | 2F/4M (66.67%) | ND | ND | ND | 1.35 (0.44–3.5) | 1.5 (0.57–14.88) |
| DTH$^+$ (n = 6) | 36 (15–56) | 5F/4M (44.50%) | ND | ND | ND | 1.03 (0.66–1.22) | 1.7 (1.17–3.6) |

Results are expressed in Median (min-max); F – female; M – Male; ND – not done; UND – undetectable (<40 viral copies/mL); rOD – relative OD.

compared with 40% (4/10) of Asympt HIV/Leish subjects and 84% (11/14) of AIDS/VL subjects. Chi-square analysis revealed that both the incorrect use of ART therapy (Fisher exact test $p = 0.0008$) and a history of previous infection with *Toxoplasma*, *M tuberculosis*, or *M. leprae* (Fisher exact test $p = 0.0007$) were associated with an increased risk of AIDS/VL outcome.

## Asymptomatic HIV/*Leishmania infantum* infection increased activation and senescence immunological status in CD4+ and CD8+T cells

T cell activation and senescence were examined in groups as listed in Table 1. The overall results are shown in S3–S9 Tables and Figs 1 and 2. CD38 and HLA-DR are markers of cell activation and high levels of these markers were observed in subjects with VL, AIDS/VL and Asympt HIV/Leish (Fig 1 and S3 Table). Subjects with AIDS/VL had more CD3+CD4+CD38+HLA-DR+ activated T cells when compared to recovered VL ($p = 0.0004$), HIV ($p = 0.0001$), and DTH+ ($p<0.0001$) (S4 Table). AIDS/VL and VL subjects had similar higher expression of activation markers, in both CD4+ and in CD8+ T cells. Recovered VL had the proportion of CD8+CD38+HLA+-DR+ activated T cells less than VL subjects. There were no differences between T-cell activation in UC subjects compared to HIV subjects, recovered VL or DTH+ subjects, in CD4+ and in CD8+ T cells subtypes. VL patients had significantly higher proportions of CD38+HLA-DR+ T cells than UC. AIDS/VL had higher proportions of activated T cells than HIV or UC (Fig 1 and S3 Table). Asympt HIV/Leish subjects had higher percentage of activated CD8+ T cells than UC subjects (Fig 1 and Table S3). Asympt HIV/Leish subjects also showed higher proportions of activated CD4+ T cells than HIV or DTH+ (Fig 1 and S3 and S4 Tables). Asympt HIV/Leish and AIDS/VL subjects had similar proportions of T cells expressing activation markers in CD4+ or CD8+ T cells. However, proportions of activated CD8+ T cells were higher in Asympt HIV/Leish subjects compared to HIV or DTH+ (Fig 1 and S1 and S5 Tables).

CD57 is one of the known senescence associated molecules. The HIV group showed significantly lower proportions of senescent CD4+T cells than Asympt HIV/Leish or AIDS/VL subjects. Nonetheless there were significantly lower proportions of senescent CD4+T cells than in Asympt HIV/Leish or in AIDS/VL subjects (Fig 2 and S3 and S6 Tables). For senescent CD8+T cells, Asympt HIV/Leish expressed higher proportions of CD57+CD8+ T cells than UC, HIV, VL and DTH+ groups (Fig 2 and S1 Table). After cure, recovered VL subjects showed similar levels of senescent CD8+ T cells compared to VL patients (Fig 2 and S3 and S7 Tables).

PD-1 (CD279) is associated with activation and exhaustion [32]. Proportions of CD4+T expressing PD1+ in AIDS/VL subjects were significantly higher than among HIV subjects (Fig 2 and S3 and S8 Table). In contrast to CD4+ T cells, the overall proportions of CD8+T cells expressing PD1+ differed significantly across the groups (Fig 2 and S3 and S9 Tables). Proportions were highest in the Asympt HIV/Leish group, which differed from proportions in HIV, DTH+ or UC subjects. Levels of CD8+ PD1+ T cells were also high in HIV, AIDS/VL and VL subjects. There was no difference in AIDS/VL in proportions of CD8+ PD1+ cells compared to VL or Asympt HIV/Leish subjects. However, AIDS/VL had higher proportions than HIV or UC. VL patients had higher expression of PD in CD8+T cells than DTH+ or UC. After cure of VL (recovered VL), the levels of CD8+PD1+ T cells were similar to those in UC and DTH+ subjects (Fig 2 and S3 and S9 Tables).

To determine which factors were associated with activation and senescence in CD4+ and CD8+T cells a multivariable analysis was performed considering *L. infantum* or HIV presence or absence, viral load, CD4+ and CD8+ counts, rOD SLA and rODrK39. The presence of *L. infantum* infection was associated with activation of CD4+T ($p = 0.04$), CD8+PD1+ ($p = 0.01$), and senescence CD8+T cells ($p = 0.006$). HIV infection was associated CD8+PD1+ T cells ($p$

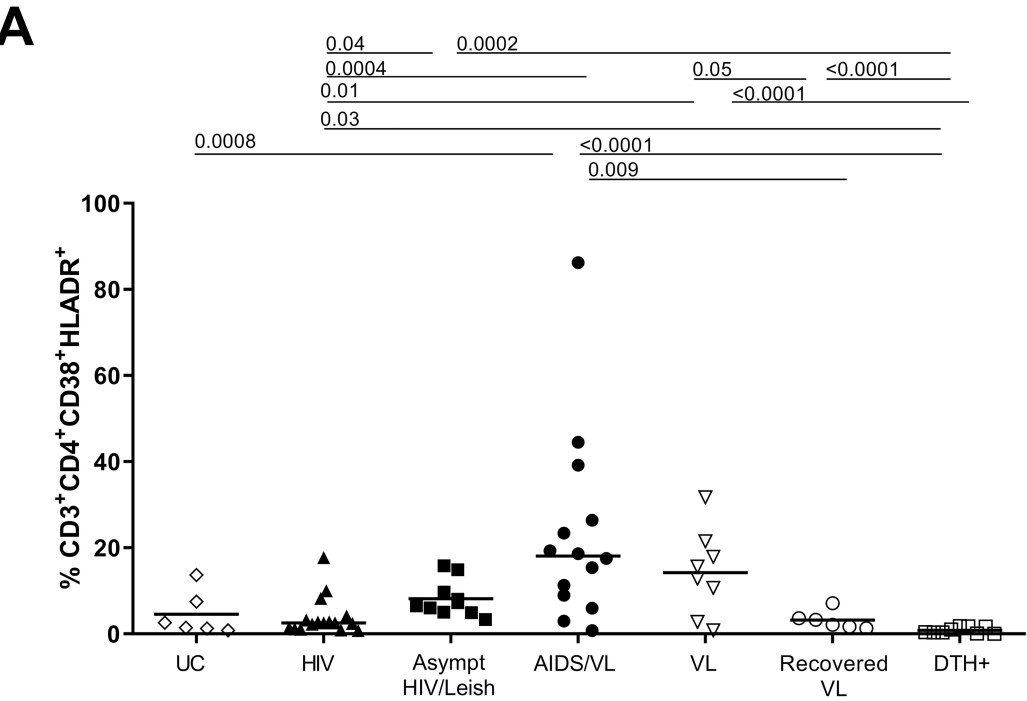

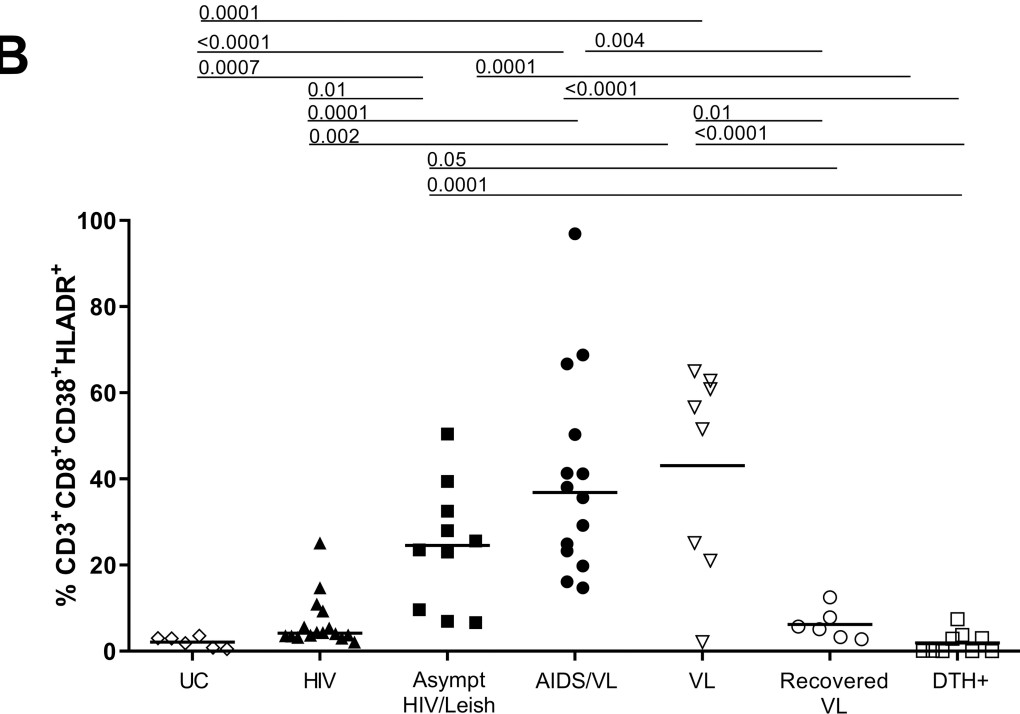

**Fig 1. Immune activation CD38+ HLA-DR+ status in CD3+CD4+ (A) and in CD3+CD8+ T cells (B).** Each point in the graphs represents one individual and horizontal bars represent median. ANOVA Kruskal Wallis test, with Dunn´s

post-test was performed with p<0.05 of significance. UC = Uninfected Controls; HIV = PLHIV negative serology to *Leishmania infantum* antigens; Asympt HIV/Leish = PLHIV with positive serology to *Leishmania infantum* antigens; AIDS/VL = subjects in AIDS and active visceral leishmaniasis; VL = active visceral leishmaniasis patients without HIV; Recovered VL = subjects without HIV after cure of visceral leishmaniasis; DTH+ = subjects without HIV with positive Montenegro skin test.

= 0.04). CD8+PD1+T cells (*p* = 0.02) or senescent CD8+T cells (*p* = 0.0008) had weak associations with CD8 counts. Spearman correlation test showed negative correlation of CD4+ counts with activated CD4+T cells (r = -0.45; p = 0.02) and with CD4+PD1+ cells (r = -0.42; p = 0.02). SLA rOD was positively correlated with senescent CD8+T cells and (r = 0.42; p = 0.03) and CD8+PD1 cells (r = 0.51; p = 0.008). All the correlations are shown in S1 Fig.

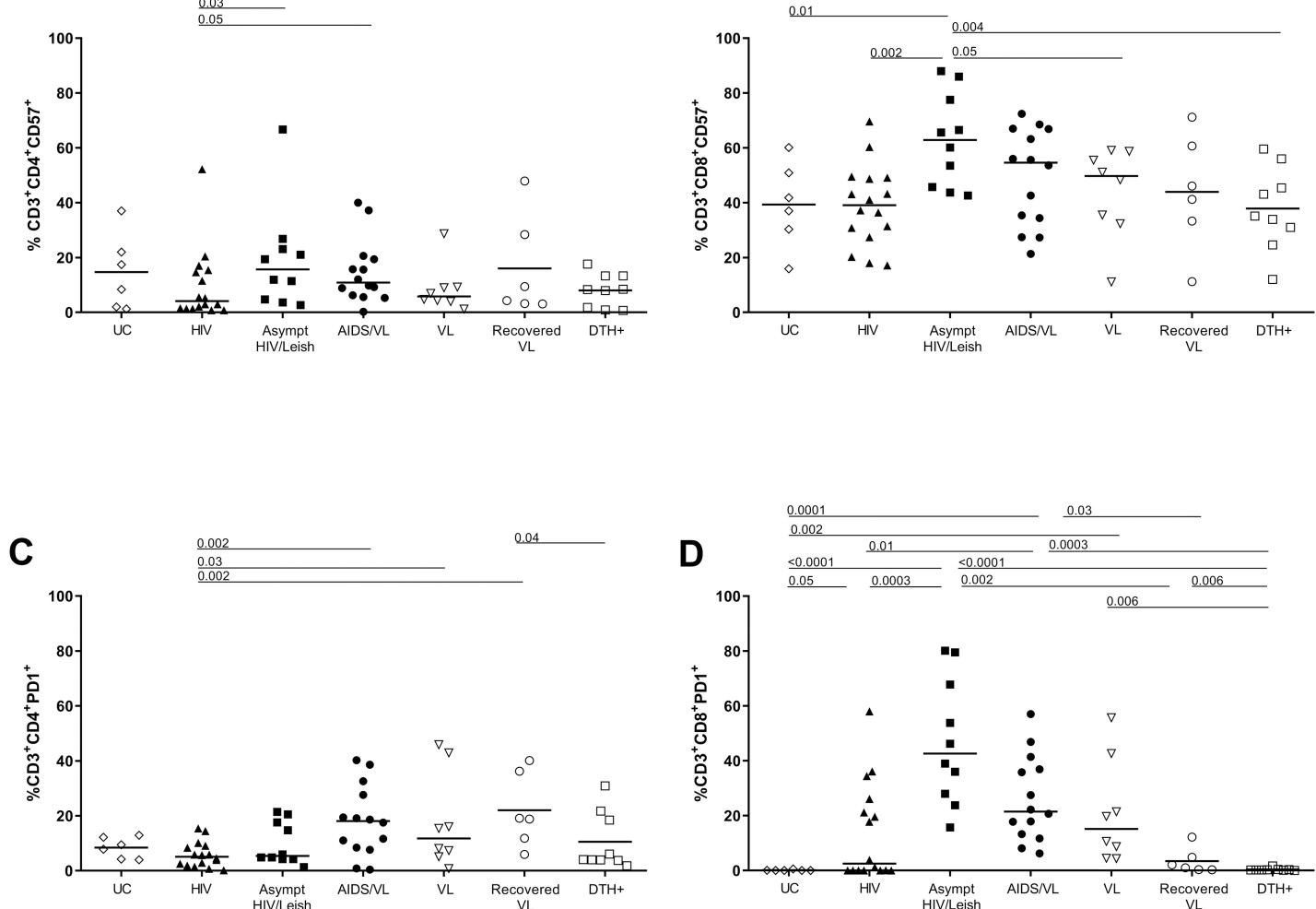

**Fig 2. Proportions of cells positive for markers of senescence (CD57+) and exhaustion (PD1+) in CD3+CD4+ (A and C) or in CD3+CD8+ T cells (B and D).** Each point in the graphs represents one individual and horizontal bars represent median. Kruskal Wallis test, with Dunn´s post-test was performed with p<0.05 of significance. UC = Uninfected Controls; HIV = PLHIV with negative serology to *Leishmania infantum* antigens; Asympt HIV/Leish = PLHIV with positive serology to *Leishmania infantum* antigens; AIDS/VL = subjects in AIDS and VL = active visceral leishmaniasis patients without HIV; Recovered VL = subjects without HIV after cure of visceral leishmaniasis; DTH+ = subjects without HIV with positive Montenegro skin test.

Principal component analysis (PCA) was performed to tease out how the concomitant infections by *L. infantum* and HIV influenced the immune response and to narrow down the variables potentially involved with the phenotypes. All 69 subjects included with the immunological parameters were considered. The parameters are the ones presented in Figs 1 and 2 and Tables 2 and S3.

The PCA analysis suggested presence of two groups: one with active *L. infantum* infection (Asymptomatic HIV/VL, AIDS/VL and VL) and a second one which included the control group (UC), HIV or people who had prior *L. infantum* infection (DTH+, Recovered VL), as shown in Fig 3. Asymptomatic HIV/VL, AIDS/VL and VL had similar patterns, as shown previously in this study. The groups with anti-Leishmania antibodies (positive serology to SLA and rK39) are people with current *L. infantum* infection by virtue of presenting with anti-Leishmania antibodies, in *L. infantum* infection antibodies may be transient and they tend to inverse correlate with a positive cellular response and may be a sign of presence of parasite. Subjects who were DTH+ or Recovered VL followed apart from groups with active Leishmania infection. HIV and UC groups, without antibody production, were grouped together with DTH+ and Recovered VL in the PCA analysis. Thus, PCA analysis suggested division of active infection in Asymptomatic HIV/VL, AIDS/VL and VL and past and/or no Leishmania infection in DTH+, Recovered VL, HIV and UC groups. As the PC scores were used to evaluate the presence of clusters in groups variables,

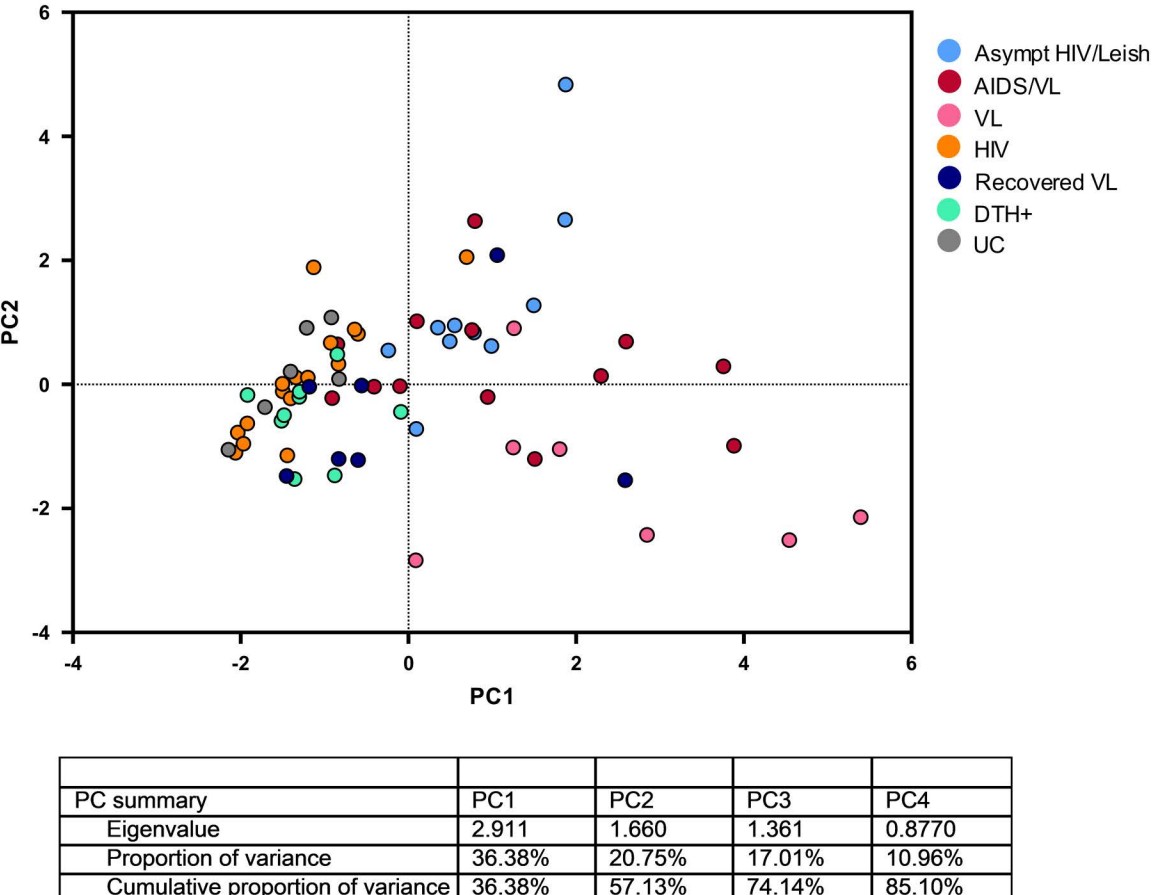

| PC summary | PC1 | PC2 | PC3 | PC4 |
|---|---|---|---|---|
| Eigenvalue | 2.911 | 1.660 | 1.361 | 0.8770 |
| Proportion of variance | 36.38% | 20.75% | 17.01% | 10.96% |
| Cumulative proportion of variance | 36.38% | 57.13% | 74.14% | 85.10% |

**Fig 3. Score plot of the principal component analysis results for immunological data for all analyzed groups.** Each dot in the graphs represents one individual. Each color in the PCA graph represents one groups of subjects: Blue dot = Asymptomatic HIV/Leish; Red dot = AIDS/VL; Pink dot = VL; Orange dot = HIV; Purple dot = Recovered VL; Green dot = DTH+; Gray dot = UC.

using K-means method, the silhouette plot showed the existence of clustering (S2 Fig), corroborating with the PCA. The K-means method suggested the existence of four clusters' groups, mainly formed by VL (group 1); AIDS/VL (group 2); DTH+, HIV, REC VL and UC (group 3) and Asympt HIV/Leish (group 4). AIDS/VL and VL cases did not discriminate perfectly and it could be related to a low number of cases (S3 Table and S2 Fig).

## Discussion

The urbanization of *L. infantum* infections in northeastern Brazil and the coincident spread of HIV have contributed to the likelihood of concurrent infections. Since 2010, an increase in AIDS/VL cases as well as in asymptomatic *L. infantum* infection have been observed in the outskirts of Natal, as shown in this study, and in other Brazilian cities [33,34]. As it is well recognized, *L. infantum* infections may yield several outcomes varying from asymptomatic to severe systemic disease, which can be fatal even with treatment. However, the majority of individuals control infection if they have intact immune system [35]. Asymptomatic infection can be detected either by a Montenegro skin test (DTH+ subjects) or more recently by a Quantiferon test, that measures IFN-ɣ production by *Leishmania*-specific cells in a research setting [36]. Peripheral blood mononuclear cells (PBMCs) from subjects with symptomatic VL have suppressed *Leishmania antigen*-specific immune responses which recover after cure, as shown by the magnitude of antigen-specific IFN-ɣ, TNF, IL-2 and IL-6. In contrast, PBMCs from patients with VL release high amounts of IL-10 [5,37]. AIDS/VL subjects have been shown to have worse prognoses for VL, with higher rate of relapses [12]. People with HIV, particularly AIDS, have lower antibodies production by new infection, thus, AIDS/VL cases produce lower levels of SLA or rK39 antibodies [34,38]. Asymptomatic HIV/*L. infantum* infection may be detected by screening HIV subjects with no symptoms of VL with different diagnostic methods to detect *L. infantum* [34].

Herein we report a high prevalence (24.2%) of *L. infantum* positive serology among residents of the state of Rio Grande do Norte, Brazil, living with HIV. The prevalence of asymptomatic *Leishmania* spp. infections in people with HIV has been reported in other VL-endemic areas at rates between 3% and 25% [39–41]. Differences in the sensitivity and specificity of *L. infantum* detection tests can affect the calculated prevalence of asymptomatic *Leishmania* spp. co-infection [42]. In the past, cured *L. infantum* infection was estimated by the Montenegro skin test, which was generated from *L. amazonensis* antigen in Brazil. Despite the difference between species, the Montenegro test was a reliable method to detect cellular T cell responses to prior *L. infantum* infection in endemic regions [20,22]. Regulatory considerations have necessitated replacing the Montenegro with Quantiferon; the overlap between the two has yet to be documented but likely they are reasonably close. However, there is no Leishmania Quantiferon clinical test available, only in research settings. In the current study, rK39 and SLA ELISA were used to screen the PLHIV for current or prior infection with *L. infantum*. Although rK39 has been reported as more sensitive for acute as opposed to prior leishmaniasis, we observed a strong correlation between SLA rOD and rK39 rOD, indicating similar humoral responses in the HIV+ population. In immunocompetent subjects, a higher rK39 ELISA was associated with VL [18]. In the population of PLHIV, either SLA or rK39 could be used to identify PLHIV with asymptomatic *L. infantum* infection. AIDS/VL patients reportedly had lower titers or no response to anti-*L. infantum* antibodies prior to the availability of ART [38]. Of note, asymptomatic HIV/Leish (PLHIV with no clinical signs of *L. infantum* infection but positive serology) have an increased risk of developing AIDS/VL. The lower rOD results in ELISA tests observed here for recovered VL and for DTH+ subjects in comparison to VL cases are similar to our prior report [18].

Asymptomatic HIV/Leish and AIDS/VL subjects in this study were predominantly males. The prevalence of HIV is higher among males than females in the state of Grande do Norte

[38]. Furthermore, symptomatic VL has also been reported to be more frequent in males [8,43]; despite the fact that both sexes were shown to be equally exposed to *L. infantum* infection in the region [20]. Demographic data suggest that sex hormones may play a role in the pathogenesis and progression to VL, once males reach puberty. Consistent with this hypothesis, higher levels of IL10 and TNF are observed in male than female mice infected with *L. infantum* [44]. Although we cannot dissect the contribution of higher male burden of HIV versus higher male susceptibility to VL in our population, it is possible that both contribute to the higher incidence of coinfection amongst males.

The consequence of HIV and *L. infantum* co-infection is an increased risk of progression to AIDS/VL according to the current and to published studies [1]. Specific T cell immune responses to *Leishmania* spp. antigens as detected by a positive DTH+ responses are a marker of protective immunity against symptomatic VL or relapsing from the disease. After cure, individuals who have recovered from VL had similar humoral and cellular immune responses as asymptomatic DTH+ subjects [5]. In the current study we observed an inverse correlation between CD4+ T cells counts and HIV viral load. Furthermore, viral loads were higher and CD4+ T cell counts were lower in AIDS/VL versus asymptomatic coinfection groups. This leads us to hypothesize there is a spectrum from asymptomatic HIV/Leish to AIDS/VL. We cannot tell from these data whether the two infections are synergistic, or whether VL symptoms are a consequence of worsening HIV infection status. To better understand that, we would need an additional group with Leishmania positive serology without HIV infection. Either way, the lack of adherence to ART therapy or a history of prior opportunistic infections were both associated with an increased risk of progressing to AIDS/VL. One assumes that adherence to ART could maintain protective Th1 responses by CD4+T cells [45].

*L. infantum* and HIV infections, alone or in presence of co-infections, cause several immunological alterations contributing to immune suppression either temporary as in the case of *L. infantum,* or long-term as in the case of HIV. In HIV infection, the co-expression of CD38 and HLA-DR on T cells was shown to predict HIV progression and correlated with high levels of viral load [1]. Persons with symptomatic VL have previously been shown to have higher percentages of *ex vivo* activation, but after *L. infantum* specific stimulation, T cell activation did not upregulate activation markers [5]. Our study showed lower *ex vivo* activation of both CD4+ and CD8+ T cells in DTH+ or recovered-VL subjects than those with Asympt HIV/Leish or AIDS/VL, although the former groups have been shown to mount specific T cell response after stimulation [5]. The results shown herein indicated high activation levels in AIDS/VL and Asympt HIV/Leish co-infected groups. The presence of *L. infantum* infection was associated with the activated status of both CD4+ and CD8+ T cells, as has been shown previously [23,46]. The chronic stimulation of cells by *Leishmania* spp. antigens could increase the levels of activation in comparison to healthy subjects. The data showed in this study suggest the presence of *L. infantum* infection may increase the levels of activation markers. In addition to direct effects of the parasite, patients with symptomatic VL have high circulating levels of LPS, sCD14 and intestinal fatty acid binding protein (-IFABP), indicating bacterial translocation across the gut mucosa may occur [6,23]. This is similar to HIV alone [47]. Nonetheless in our subjects, co-infection with *L. infantum* and HIV, in both the HIV/Leish and AIDS/VL groups, was associated with higher T cell activation markers in CD4+T and CD8+ T cells. The activation status of CD4+T and CD8+T cells was similar in both coinfected groups.

PD-1 expression is induced by T cell activation, and in a response to a strong TCR signal, it can suppress T cell functions. Cells with exhausted phenotype express the markers PD-1, TIM-3, LGA-3 [48,49]. In active symptomatic canine VL, exhausted CD4+ and CD8 T cells, increase PD-1 expression, decrease cell proliferation and IFN-γ production [46]. Chronic T activation, such as

occurs in HIV infection or VL, can lead to dysfunctional T cells with reduced responses to antigen stimulation. Higher percentages of CD4[+] or CD8[+]T cells expressing markers CD57[+] (senescence) or PD1[+] (marker of activation or exhaustion) were observed in asymptomatic HIV/Leish and AIDS/VL subjects compared to other groups. Senescent CD8[+]T (CD57[+]) or CD8[+]T PD1[+] were similar and higher between asymptomatic HIV/Leish, AIDS/VL and VL subjects. A multivariate analysis indicated that *L. infantum* infection was associated with the levels of senescent CD8[+] or CD8[+] PD1[+] cells. Senescent cells have shortened telomeres that can affect cellular function and proliferation [32]. Exhausted or senescent cells lost the capacity to perform their functions [50], as showed by reduced T cell proliferation and cytokine production after *Leishmania* antigen stimulation [5,37]. Exhausted or anergic CD8[+]T cells from active VL patients did not produce IFN-γ in response to specific stimulation. However, this suppression was transient, since after treatment, CD8[+]T cells recover the capacity of specific response [51]. Senescence of T cells was more accentuated in AIDS/VL than in HIV subjects in our study, although CD4[+] T cell counts were significantly higher in the HIV group. Asymptomatic HIV/Leish T cells displayed similar proportions of senescence markers compared to the AIDS/VL group, raising the possibility that asymptomatic HIV/Leish group could be in the beginning stages of progression to functional impairment. Moreover, this similarity between Asymptomatic HIV/Leish and HIV/Leish group may be due to chronic HIV and/or *Leishmania infantum* antigen exposure or to increase of gut barrier permeability.

Asymptomatic HIV/Leish co-infected subjects seemed to be controlling both infections: this group did not have any VL symptoms and had similar HIV viral loads, CD4[+] and CD8[+] T cell counts in comparison to HIV subjects. These similarities reflect other published reports [39]. Based on our observations, we hypothesize that progression of asymptomatic to symptomatic VL in PLHIV, could be result of lack immune control of HIV infection, cause by poor ART adherence. The use of ART decreased the symptomatic VL in PLHIV as observed before [22]. As observed in HIV/Leish co-infection, the active *L. infantum* infection stimulates immune activation [23], leading to increased HIV viral load and accelerating progression to AIDS/VL [12,52]. Our results showed a greater risk of developing AIDS/VL in asympt HIV/Leish subjects who had low adherence to treatment and had previous infections. Together the above results suggest that *L. infantum* infection may be able to worsen caused by HIV infection, elevating the levels of activated and senescent T cells and amplifying disease progression.

This work had some limitations: 1) all PLHIV were studied in an era when ART was deferred until a relatively lower CD4 cell count, and some individuals had uncontrolled HIV and histories of several infectious complications. Moreover, we did not know when the subjects acquired HIV infection. 2) another limitation of this work was that we did not do PCR or DTH testing for *L. infantum* to assess the asymptomatic HIV/Leish group, and only positive serology was considered. 3) We did not include an additional group of subjects with positive Leishmania serology without HIV; instead, we evaluated DTH+ subjects and this group was considered immunocompetent asymptomatic VL; 4) Other markers as Tim-3, Lag-3 were not tested with PD-1, to determine exhaustion in T cells. In immunocompetent subjects, active VL and asymptomatic *L. infantum* infection present high levels of anti-Leishmania antibodies [53], but most AIDS/VL subjects present low levels of Leishmania antibodies and 30% are anti-Leishmania antibody negative [38]. To assess Leishmania infection in PLHIV, positive Leishmania serology was used to classify persons as asymptomatic HIV/Leishmania co-infected. We are not able to infer causality due to the cross-sectional design; thus, most of our results should be hypothesis-generating only. Furthermore, there may be significant confounding variables, such as gender, age, and ART adherence. Most HIV subjects were regularly on ART (13/16; 81.25%), while only 6 of 10 (60%) of Asympt HIV/Leish adhered to ART appropriately. ART adherence can influence the frequencies of T cell activation, senescence and exhaustion, resulting in decreased risk of co-morbidity [17].

The immunological study showed that *L. infantum* itself is associated with T cell activation and T cell senescence. However, in immunocompetent subjects this is temporary; after cure the frequencies of activated and senescent T cells decrease, as we have shown here in the Recovered VL group. In AIDS/VL subjects, the chronic activation induces loss of peripheral T cells and immune reconstitution could be achieved after HIV treatment, if the subject did not have a new episode of AIDS/VL (relapsing) [54].

Finally, the PCA showed that there is discrimination between subjects with anti-Leishmania antibodies and the remaining groups either controls or subjects with a previous history of *L. infantum* infection (DTH+). It is likely that active *L. infantum* infection results in greater activation and senescence of T cells. However, the major caveat is that those parameters decrease after resolution of the *L. infantum* infection either by treatment or spontaneously. A prior study with AIDS/VL in another population showed that there is gut bacterial translocation, which could increase the amount of LPS and induce greater inflammatory response [23].

The management of asymptomatic HIV/Leishmania co-infected subjects starts with optimizing ART therapy and close follow-up of clinical status, viral load and CD4+ T count. The role of anti-leishmanial therapy in the asymptomatic co-infected person is uncertain, although our data raise the possibility that it may be very important to maintain the infection in remission. Longitudinal trials are needed to determine whether, in the setting of universal ART and sustained viral load suppression, T cell activation and exhaustion are seen in PLHIV with positive serology for Leishmania and, if so, whether these immune parameters predict active VL despite suppressed virus.

## Supporting information

**S1 Fig. Spearman correlation of activation and exhaustion considering CD4+ counts and anti-leishmania antibodies.** A. Activation in CD4+; B. Exhaustion in CD4+, C. Senescence in CD8+ and D. Exhaustion in CD8+.
(TIF)

**S2 Fig. Silhouette plot for the K-means clustering analysis using 3 PC's obtained from immunological parameters.** For clusters 1, 2, 3, and 4, the number of subjects (the average silhouette width) in each cluster are 6 (0.58), 40 (0.76), 10 (0.32) and 12 (0.37), respectively. The best value of the average silhouette width was 0.61. s_i is the silhouette width (or score) and J is the cluster name and 1, (n_j) is number of individuals in the cluster and ave_iEC_i is the mean of the silhouette width in cluster. Each cluster has a j, n-j and ave_iEC_i.
(TIF)

**S1 Table. Pairwise comparisons of the marginal mean of the OD_SLA considering a Generalized Linear Model fit.**
(DOCX)

**S2 Table. Pairwise comparisons of the marginal means of CD4 and CD8 counts via Generalized Linear Mode fit.**
(DOCX)

**S3 Table. Proportions of subjects with makers of immune activation, senescence or exhaustion in T cell subsets.**
(DOCX)

**S4 Table. Pairwise comparisons of the marginal means of %CD3CD4CD38HLADR via GLM fit.**
(DOCX)

**S5 Table. Pairwise comparisons of the marginal means of the percent of CD3CD 8CD 38HLADR via GLM fit.**
(DOCX)

**S6 Table. Pairwise comparisons of the marginal means of %CD3CD4CD57 proportion via beta regression model fit.**
(DOCX)

**S7 Table. Pairwise comparisons of the marginal means of the percent of CD3CD8CD57 proportion via beta regression model fit.**
(DOCX)

**S8 Table. Pairwise comparisons of the marginal means of the percent of CD3CD4PD1 proportion via beta regression model fit.**
(DOCX)

**S9 Table. Pairwise comparisons of the marginal means of CD3CD8PD1 proportion via beta regression model fit.**
(DOCX)

**S10 Table. PCA loading for each analyzed variables.**
(DOCX)

**S11 Table. Distribution of individuals into K-means cluster groups.**
(DOCX)

**S1 Data. Unidentified raw serological, clinical and laboratory data.**
(XLSX)

## Acknowledgment

We are thankful to the health personnel from Giselda Trigueiro Hospital, Central Laboratory (Laboratório Central de Saúde Pública do Rio Grande do Norte Dr. Almino Fernandes) for helping with the volunteers' recruitment.

## Author contributions

**Conceptualization:** Carolina de Oliveira Mendes-Aguiar, Eliana Lúcia Tomaz Nascimento, Selma Maria Bezerra Jerônimo.

**Data curation:** Jose Wilton Queiroz, Eliardo G Costa, Selma Maria Bezerra Jerônimo.

**Formal analysis:** Jose Wilton Queiroz, Eliardo G Costa, Marshall J Glesby.

**Funding acquisition:** Selma Maria Bezerra Jerônimo.

**Investigation:** Carolina de Oliveira Mendes-Aguiar, Manoella Monte Alves, Amanda de Albuquerque Lopes Machado, Glória Regina Gois Monteiro, Iara Marques Medeiros, Iraci Duarte Lima, Eliana Lúcia Tomaz Nascimento, Selma Maria Bezerra Jerônimo.

**Methodology:** Carolina de Oliveira Mendes-Aguiar, Amanda de Albuquerque Lopes Machado, Glória Regina Gois Monteiro, Iara Marques Medeiros, Richard D. Pearson, Mary E Wilson, Marshall J Glesby, Selma Maria Bezerra Jerônimo.

**Project administration:** Selma Maria Bezerra Jerônimo.

**Resources:** Manoella Monte Alves, Glória Regina Gois Monteiro, Iara Marques Medeiros, Iraci Duarte Lima, Selma Maria Bezerra Jerônimo.

**Supervision:** Selma Maria Bezerra Jerônimo.

**Writing – original draft:** Carolina de Oliveira Mendes-Aguiar.

**Writing – review & editing:** Carolina de Oliveira Mendes-Aguiar, Jose Wilton Queiroz, Eliardo G Costa, Richard D. Pearson, Mary E Wilson, Marshall J Glesby, Eliana Lúcia Tomaz Nascimento, Selma Maria Bezerra Jerônimo.

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
