## [Decision Letter · Decision Letter 0]

9 May 2024

Dear Dr. Jeronimo,

Thank you very much for submitting your manuscript "T-cell activation, senescence, and exhaustion in asymptomatic HIV/Leishmania infantum co-infection" for consideration at PLOS Neglected Tropical Diseases. As with all papers reviewed by the journal, your manuscript was reviewed by members of the editorial board and by several independent reviewers. In light of the reviews (below this email), we would like to invite the resubmission of a significantly-revised version that takes into account the reviewers' comments. 

Comments Reviewer 2:

This manuscript characterized the prevalence of Leishmania infantum antibodies and delayed type hypsersensitivity reactions in a cross-sectional study of 1,372 people living with HIV infection in NE Brazil between 2014 and 2016. As visceral leishmaniasis (VL) is a reportable disease, VL cases occurring between 2014 and 2018 were cross-indexed with the cohort to identify cases. 69 subjects with HIV and/or L. infantum from the cross-sectional study were recruited for additional immunological evaluations. As described in other reports, the prevalence of infection with L. infantum was high based on antibody testing (ELISA against two antigens) and DTH responses. Different clinical outcomes of L. infantum (asymptomatic infection in HIV+ and HIV- subjects, VL with AIDS, symptomatic VL, recovered VL (following treatment), DTH+ with no history of VL, or healthy control. Flow cytometry studies examined immune activation (CD38+/HLADR+), senescence (CD57+), and exhaustion (PD-1)

markers in CD4+ and CD8+ T cells. The VL follow up in the cross-sectional cohort found that a positive ELISA was associated with a 2.27 relative risk of developing AIDS/VL over a relatively short period of follow up (< 4 years). This is important information that suggests that L. infantum serologic testing should be strongly considered in people living with HIV in endemic regions.

More information regarding the group recruited for immunological studies is needed and the definition of the groups should be clarified. It is stated the 69 subjects were “drawn from the cross-sectional study” (line 169), thus one would assume all are HIV+. However, there are 6 HIV negative control subjects, and 17 with VL that appear to not have HIV infection (8 VL and 9 recovered VL). Were these individual’s part of the original cross-sectional study?

The study lacks important information on HIV infection and treatment that may greatly influence the activation marker data interpretation. Specifically, AIDS/VL had uncontrolled HIV RNA with the known association of high viral load and reduced CD4+ T cell counts, increased activation, senescence and increased exhaustion markers. Since these activation markers are elevated in the VL and recovered VL groups (presumably HIV-negative), it is logical to assume that VL enhances the chronic immune activation associated with HIV infection. Since immune activation also increases HIV viral load, and HIV viral load is inversely related to CD4 counts, it is possible that the VL is partially responsible for the worsened HIV markers suggesting a synergistic relationship, although the number of subjects is low and the power to dissect this relationship is limited. The assumption by the authors that the high HIV viral load was due to poor antiretroviral therapy (ART) adherence.

Specifically, how do the authors know that this is due to poor adherence versus newly diagnosed HIV infection? More information about the timing of ART in these individuals and how it relates to the immunologic results is needed to interpret these data. If HIV+ people with VL present for care before or shortly after they receive ART, this confounds all activation measurements as there are HIV-mediated effects and immune reconstitution effects that may contribute to the activation, senescence and exhaustion results.

In summary, this manuscript provides data on the risk of VL in an HIV cohort based on retrospective reportable disease data. The paper further describes immunological evaluations in a subgroup of this cohort. Better definition of the group studied, and description of the HIV therapy in this group is needed to interpret the T cell activation/senescence/exhaustion findings.

We cannot make any decision about publication until we have seen the revised manuscript and your response to the reviewers' comments. Your revised manuscript is also likely to be sent to reviewers for further evaluation.

Sincerely,

Johan Van Weyenbergh

Academic Editor

Susan Madison-Antenucci

Section Editor

Comments Reviewer 2:

This manuscript characterized the prevalence of Leishmania infantum antibodies and delayed type hypsersensitivity reactions in a cross-sectional study of 1,372 people living with HIV infection in NE Brazil between 2014 and 2016. As visceral leishmaniasis (VL) is a reportable disease, VL cases occurring between 2014 and 2018 were cross-indexed with the cohort to identify cases. 69 subjects with HIV and/or L. infantum from the cross-sectional study were recruited for additional immunological evaluations. As described in other reports, the prevalence of infection with L. infantum was high based on antibody testing (ELISA against two antigens) and DTH responses. Different clinical outcomes of L. infantum (asymptomatic infection in HIV+ and HIV- subjects, VL with AIDS, symptomatic VL, recovered VL (following treatment), DTH+ with no history of VL, or healthy control. Flow cytometry studies examined immune activation (CD38+/HLADR+), senescence (CD57+), and exhaustion (PD-1)

markers in CD4+ and CD8+ T cells. The VL follow up in the cross-sectional cohort found that a positive ELISA was associated with a 2.27 relative risk of developing AIDS/VL over a relatively short period of follow up (< 4 years). This is important information that suggests that L. infantum serologic testing should be strongly considered in people living with HIV in endemic regions.

More information regarding the group recruited for immunological studies is needed and the definition of the groups should be clarified. It is stated the 69 subjects were “drawn from the cross-sectional study” (line 169), thus one would assume all are HIV+. However, there are 6 HIV negative control subjects, and 17 with VL that appear to not have HIV infection (8 VL and 9 recovered VL). Were these individual’s part of the original cross-sectional study?

The study lacks important information on HIV infection and treatment that may greatly influence the activation marker data interpretation. Specifically, AIDS/VL had uncontrolled HIV RNA with the known association of high viral load and reduced CD4+ T cell counts, increased activation, senescence and increased exhaustion markers. Since these activation markers are elevated in the VL and recovered VL groups (presumably HIV-negative), it is logical to assume that VL enhances the chronic immune activation associated with HIV infection. Since immune activation also increases HIV viral load, and HIV viral load is inversely related to CD4 counts, it is possible that the VL is partially responsible for the worsened HIV markers suggesting a synergistic relationship, although the number of subjects is low and the power to dissect this relationship is limited. The assumption by the authors that the high HIV viral load was due to poor antiretroviral therapy (ART) adherence.

Specifically, how do the authors know that this is due to poor adherence versus newly diagnosed HIV infection? More information about the timing of ART in these individuals and how it relates to the immunologic results is needed to interpret these data. If HIV+ people with VL present for care before or shortly after they receive ART, this confounds all activation measurements as there are HIV-mediated effects and immune reconstitution effects that may contribute to the activation, senescence and exhaustion results.

In summary, this manuscript provides data on the risk of VL in an HIV cohort based on retrospective reportable disease data. The paper further describes immunological evaluations in a subgroup of this cohort. Better definition of the group studied, and description of the HIV therapy in this group is needed to interpret the T cell activation/senescence/exhaustion findings.

Reviewer's Responses to Questions

<b>Key Review Criteria Required for Acceptance?

**Methods**

-Are the objectives of the study clearly articulated with a clear testable hypothesis stated?

-Is the study design appropriate to address the stated objectives?

-Is the population clearly described and appropriate for the hypothesis being tested?

-Is the sample size sufficient to ensure adequate power to address the hypothesis being tested?

-Were correct statistical analysis used to support conclusions?

-Are there concerns about ethical or regulatory requirements being met?

Reviewer #1: (No Response)

Reviewer #2: (No Response)

Reviewer #3: - Are the objectives of the study clearly articulated with a clear testable hypothesis stated?

The hypothesis was that people with coinfection with HIV and L. infantum would have altered T cell activation which would increase risk of visceral leishmaniasis. The authors should specify coinfection relative to what – presumably should be relative to L. infantum mono-infection. The presence of altered T cell activation could be (and was) easily tested but whether this increased activation would increase risk of visceral leishmaniasis is not easy to test.

- Is the study design appropriate to address the stated objectives?

The study design is appropriate to test the first part of the hypothesis but not the second part. For the second part, the authors have compared immune profiles of participants with asymptomatic Leishmania infection plus HIV with those of symptomatic Leishmania plus HIV. However, it seems that this was a cross-sectional rather than longitudinal analysis so no causality can be attributed to the findings. To answer the question whether T cell activation influences the development of leishmaniasis, you would really have to do a longitudinal analysis of the T cell immunology of people with HIV and asymptomatic leishmaniasis and compare the immunology of those who develop symptomatic leishmaniasis versus those who don’t. This would have to be a large study considering that only 2.4% of participants with asymptomatic coinfection developed symptoms over three years. Even this would not be definitive – you would then need to design a trial where you somehow specifically reduced T cell activation to see whether this had an effect on development of symptomatic leishmaniasis. This would also have to be a very large study.

Considering the difficulty in funding such large studies, and the presumed paucity of evidence in this field, the authors have done a cross-sectional analysis which at least supports their hypothesis even if it cannot confirm it. The authors should highlight these issues and limitations in their discussion.

One other limitation is the variable rate of HIV viraemia and adherence to ART in the study population as seen in figure 2 and table 4. This is likely a major confounder for the association between T cell activation and development of asymptomatic and symptomatic leishmaniasis. The authors did not measure antigen-specific T cell activation so this may have been non-specific as a consequence of viraemia rather than a response to L. infantum. Indeed, the authors highlight that antigen-specific T cell activation is minimal in active leishmaniasis. They argue that asymptomatic coinfected participants likely had antigen-specific T cell responses (lines 450-454) but this is on the basis of weak evidence and they should acknowledge that it may alternatively just relate to reduced levels of HIV suppression. Indeed, they found that adherence to ART was highly predictive of the clinical leishmania status in people living with HIV. The authors should report any correlation between HIV viral load and T cell activation which they stated they performed in lines 361-364.

This distinction matters because it affects interpretation of the results. It may simply be that maintenance of suppressive ART is all that is needed to prevent either asymptomatic or symptomatic leishmania infection. The authors argue that there may be a role for anti-leishmania treatment in asymptomatic coinfected individuals but they need to acknowledge that without antigen-specific testing they cannot determine whether the T cell activation observed in these individuals is helping to control infection. They should also acknowledge that, with rates of development of symptomatic leishmaniasis of only 2.4% within three years amongst those with initially asymptomatic coinfection, it would be necessary through longitudinal studies to identify risk factors within this group for development of active disease in order to stratify which asymptomatic coinfected patients might benefit from anti-leishmania treatment. This could then be tested in a clinical trial.

- Is the population clearly described and appropriate for the hypothesis being tested?

It isn’t entirely clear where some of the 69 participants for the immunologic analyses came from. They can’t all have come from the cross-sectional analysis, as stated in lines 154-155, because the cross-sectional study was only of people living with HIV who presumably were all asymptomatic of leishmaniasis when they had their initial serology. Were the AIDS/VL participants all initially part of the cross-sectional study? This should be stated. Also, where did the HIV-negative participants come from?

Please note the correct term is 'people living with HIV', not 'HIV positive subjects', ie refer to the person first and the disease second. This is globally accepted as appropriate terminology as requested by numerous advocacy groups.

- Is the sample size sufficient to ensure adequate power to address the hypothesis being tested?

Yes it clearly was because the results were significant. Trying to assess this prospectively is nothing more than a guessing game if there is no preliminary data – this paper now provides preliminary data for future studies (at least in this region).

- Were correct statistical analysis used to support conclusions?

I believe so. However, it appears that no statistician was used to analyse results; the authors instead used GraphPad Prism and ‘omni calculator’ available online. It is probably worth getting a statistician to review the analytical methods.

- Are there concerns about ethical or regulatory requirements being met?

The authors should state whether the consent form was signed before any study-related activities were commenced. They should also state whether the study was registered online prior to commencement.

**Results**

-Does the analysis presented match the analysis plan?

-Are the results clearly and completely presented?

-Are the figures (Tables, Images) of sufficient quality for clarity?

Reviewer #1: (No Response)

Reviewer #2: (No Response)

Reviewer #3: - Does the analysis presented match the analysis plan?

Not entirely; the authors should mention the correlation analyses – which ones were prospectively decided on and how the correlation analyses were performed.

- Are the results clearly and completely presented?

I don’t understand why tables S1 to S4 were included when this information is already in figures 1 to 4 (apart from the non-significant p values – which can just be omitted as it is clear they are non-significant, ie p > 0.05). There are some discrepancies – eg table S2 AIDS/VL vs asymptomatic HIV/leish, table S3 CD57 vs recovered VL and PD-1+ vs DTH+ (both for CD4+ T cells) – please check whether the table or the figure is correct. I also don’t understand the purpose of table 2 – all of this information is already in the text.

Line 220 – what does ‘high titers’ mean? Suggest changing this to ‘positive titers’ if appropriate. Line 227 – should reword to say relative risk for development of VL in participants living with HIV with versus without positive leishmania serology (ie specify what this is relative to). Line 239 – suggest using a different acronym to CS – not clear what this stands for (presumably control subjects) – suggest UC for uninfected controls and spell this out.

Was there any significant difference between groups in gender balance? Was any age / gender matching performed between groups selected for immunologic analysis? What do the horizontal bars indicate on the graphs? This should be stated in the legends. HIV appears twice in the x axis for figure 2A.

Description of some of the results is far too wordy and difficult to read – especially lines 311 to 328. This doesn’t add anything over just looking at the figure which is far easier to interpret. I suggest simplifying the description significantly, focusing on overall and interesting results and leaving out most of the non-significant findings unless they are interesting/surprising. It can also be useful to introduce these multiple findings by starting with what you expected (and why) and then reporting what you actually found.

Why were some correlation analyses interspersed with the other results when other correlation analyses were reported in the final paragraph? I would suggest putting the correlation analyses all together in the final paragraph.

In lines 336-7 the authors state that asymptomatic HIV/Leish had more senescent CD8+ T cells than HIV but they appear to contradict this in the next sentence; I think the wording of that next sentence needs to be changed. Again, I would suggest focusing on the significant results rather than the non-significant ones. Line 341 – a positive correlation should have a positive r value. Line 351 – beware of saying ‘similar proportions’ when they are not similar at all – they are just not statistically different. Line 359 - ? missed the word rk39? Line 362 – was this really a multivariable analysis (ie multiple regression) or just multiple separate correlation analyses? The term multivariable analysis implies multiple regression.

- Are the figures (Tables, Images) of sufficient quality for clarity?

Yes.

**Conclusions**

-Are the conclusions supported by the data presented?

-Are the limitations of analysis clearly described?

-Do the authors discuss how these data can be helpful to advance our understanding of the topic under study?

-Is public health relevance addressed?

Reviewer #1: (No Response)

Reviewer #2: (No Response)

Reviewer #3: - Are the conclusions supported by the data presented?

Not entirely. Line 450 – overreach to say that it is ‘probable’ that asymptomatic HIV/Leish subjects can mount a specific Th1 response to SLA simply based on similar frequencies of activated CD4+ T cells to DTH and lower levels of CD4+ T cell activation than AIDS/VL participants.

Lines 480-483 – overreach to say ‘suggesting’ asymptomatic pts could be in the beginning stages of progression to functional impairment. The authors could perhaps say ‘raising the possibility’. Most participants didn't develop VL. It could also be that senescence / exhaustion reflects increased HIV or Leish antigen exposure or gut barrier permeability and could be adaptive in preventing disease and that other factors are responsible for developing symptoms. The authors should mention this possibility.

Line 484-485 – participants are mostly on ART in this group so it isn’t accurate to say that they seem to be mostly controlling both infections.

Lines 487-490 – a much more likely explanation is that most participants in the AIDS/VL group weren’t taking ART. I suggest starting with this and then saying that asymptomatic VL may have had an additional impact on T cell exhaustion although one can’t exclude that this was also due to being less likely on ART than those who weren’t infected with L. infantum.

Line 493 – the word ‘can’ should be changed to ‘may be able to’ and clearly defined – to refer to T cell exhaustion and senescence rather than CD4+ T cell decline.

Line 499 – ‘suggest’ is an overreach. I suggest changing this to ‘raise the possibility that’.

Lines 502-503 - This paper provides no evidence to support pre-emptive therapy. I suggest concluding that longitudinal studies are required to determine whether elevation in T cell senescence and exhaustion predicts VL even in those with suppressed virus.

Line 446 - Reference 5 shows that antigen stimulation did not UPREGULATE activation markers above baseline (compare figures 1 and 2 in reference 5), not that the cells didn’t EXPRESS activation markers following antigen stimulation.

Line 456 – This line appears to conflict with lines 445-446; the authors should explain what they mean more clearly.

- Are the limitations of analysis clearly described?

No, not at all. The authors should highlight that causality cannot be inferred from a cross-sectional study and that this paper is hypothesis-generating only. Furthermore, the lack of even gender/age matching makes it hard to interpret whether differences between groups could be due to these factors or other potential confounders. The lack of antigen-specific testing makes it hard to know whether elevations in immune activation are a response to parasite antigen or not. The difference between groups in ART adherence is likely a major confounder. They need to highlight which investigations would be required next to take things further.

- Do the authors discuss how these data can be helpful to advance our understanding of the topic under study?

No. They draw tentative conclusions without adequately acknowledging alternative more likely explanations or drawing attention to the limitations of the study.

- Is public health relevance addressed?

Not really. They highlight the surprisingly high frequency of seropositivity amongst people living with HIV and the changing epidemic with HIV entering rural areas and L. infantum entering urban areas but don’t discuss how this could be addressed from a public health perspective (eg long-sleeved clothing, insect repellent, bed nets).

**Editorial and Data Presentation Modifications?**

Reviewer #1: (No Response)

Reviewer #2: (No Response)

Reviewer #3: Please see first and third paragraphs under “Are the results clearly and completely presented?” in the results section above.

**Summary and General Comments**

Reviewer #1: Visceral leishmaniasis can be fatal, and co-infection with HIV increases the risk of death. Therefore, understanding the impact of HIV on responses to VL (and vice-versa) is important. This manuscript aims to study changes in the phenotype of T cells in co-infected patients, and the authors have access to a large number of patients to analyze the different functions of T cells. This could lead to a greater understanding of the immune response to these infections, and how they impact diseases. However, the analysis is limited, the description of the results needs to be improved, and the data representation could be more friendly to the reader. In addition, the interpretation of the data is overstated, and the results are not contextualized with the literature. 

Specific comments:

1. There is no explanation of what is T cell senescence and exhaustion and why the authors chose these functions and those markers (CD57 and PD1) to characterize them. 

2. How do the authors know that AIDS/VL subjects have VL? Is DTH the criteria? This is important because then it means that these patients have defects in antibody production, but T cell responses are unaffected. If correct, how do the authors interpret those findings? The criteria should be clearly explained in the methods section. 

3. No interpretation of the findings is provided, making it difficult to read the manuscript. For example: “a negative correlation was observed between activated CD4+T cells and CD4+ counts”. What is the authors’ interpretation of this finding? This lack of interpretation and contextualizing of the results with the literature is widespread throughout the manuscript. 

4. What is the relative risk of developing AIDS/VL compared to other infectious diseases? 

5. PD-1 is a T cell activation marker. The expression of PD-1 alone does not mean that T cells are exhausted. How do the authors interpret their findings, given that they only looked for PD-1 expression?

6. There is no consensus in the literature on whether CD4 T cells become exhausted because there is no clear demonstration that they have the loss of effector functions like CD8 T cells do. The authors make no comments on this. 

7. In lines 345-346, the authors state: “Remarkably, there were few other significant differences in CD4+ T cell exhaustion markers between the groups”. Why not explain what those small differences are and what they mean? Also, why some of the data is shown as a table instead of a figure?

8. What is the meaning of the “weak association” between exhausted and senescent CD8 T cells and CD8 counts (lines 367-368).

Minor comments:

1. Describe the acronyms in figure/table legend. CS, for example, needs to be described.

Reviewer #2: (No Response)

Reviewer #3: Please see comments already provided. Additionally, the authors should comment on whether a study of this nature has been published before (ie whether these are novel findings). Overall there are a large number of issues that should be addressed before this paper could be considered for publication.

PLOS authors have the option to publish the peer review history of their article (what does this mean? ). If published, this will include your full peer review and any attached files.

**Do you want your identity to be public for this peer review?** For information about this choice, including consent withdrawal, please see our Privacy Policy .

Reviewer #1: No

Reviewer #2: No

Reviewer #3: No
---

## [Decision Letter · Decision Letter 1]

20 Sep 2024

Dear Dr. Jeronimo,

Thank you very much for submitting your manuscript "T-cell activation, senescence, and exhaustion in asymptomatic HIV/Leishmania infantum co-infection" for consideration at PLOS Neglected Tropical Diseases. As with all papers reviewed by the journal, your manuscript was reviewed by members of the editorial board and by several independent reviewers. In light of the reviews (below this email), we would like to invite the resubmission of a significantly-revised version that takes into account the reviewers' comments. 

Please note that unless all of the reviewers comments are addressed in a substantial way it is likely that we will not be able to proceed further with this manuscript.

We cannot make any decision about publication until we have seen the revised manuscript and your response to the reviewers' comments. Your revised manuscript is also likely to be sent to reviewers for further evaluation.

Sincerely,

Johan Van Weyenbergh

Academic Editor

Susan Madison-Antenucci

Section Editor

Reviewer's Responses to Questions

**Key Review Criteria Required for Acceptance?**

**Methods**

-Are the objectives of the study clearly articulated with a clear testable hypothesis stated?

-Is the study design appropriate to address the stated objectives?

-Is the population clearly described and appropriate for the hypothesis being tested?

-Is the sample size sufficient to ensure adequate power to address the hypothesis being tested?

-Were correct statistical analysis used to support conclusions?

-Are there concerns about ethical or regulatory requirements being met?

Reviewer #1: (No Response)

Reviewer #2: The authors have addressed the concerns raised in my first review whenever possible, and the manuscript is strengthened by these changes.

Reviewer #3: The wording of this redraft is much improved. However, there is still a lot of work to do.

Abstract

- Lines 39-40 – altered T cell activation relative to whom? The authors still haven’t specified this and need to do so as otherwise this sentence is rather meaningless. Based on the content of the paper, the answer should be relative to PLHIV without Leishmaniasis. This part of the hypothesis was addressed in the paper but the second part was not (“that would increase the risk of VL”) – this part can only be evaluated with an interventional study. I would say something like “We hypothesised that, relative to those with HIV alone, people with co-infection would have altered T cell activation which could impact on the risk on the risk of development of VL”.

- I suggest not referring to asymptomatic HIV/Leish, symptomatic VL, recovered VL etc in the abstract as these terms are meaningless to readers. I suggest the authors rather describe what they mean – eg Immune status of 69 participants was evaluated and comparisons made between those with HIV without, with latent or with active Leishmania infection and those without HIV but with active or resolved Leishmania or T cell hypersensitivity to Leishmania antigen. There are multiple instances where the authors have referred to asymptomatic HIV/Leish etc which should be changed.

- The authors should include the confidence intervals around the relative risk of 2.2 and a p value given that this was one of the key questions of interest identified in the hypothesis; I presume this is not significant given that the confidence interval crosses 1. They should also explain who the comparator group is in this sentence.

- “Poor adherence to antiretroviral therapy… was associated with development of AIDS/VL” - the authors should specify which group they are referring to (?HIV alone or HIV with positive Leish serology or both).

Author summary

- PLWH – it is requested by people living with HIV to avoid turning them into an acronym. When an acronym is necessary due to word limits, the correct one is PLHIV (without a W). However, wherever possible, it is best to write “people living with HIV” rather than PLHIV. There are multiple instances of PLWH throughout the paper.

- Referring to asymptomatic HIV/Leish, symptomatic VL, recovered VL etc in the author summary is fairly meaningless to readers who haven’t read the paper yet – I would suggest not using these terms in the author summary but simply describing the comparisons which are most relevant, eg markers of ___ were higher in people living with HIV with positive Leishmania serology compared to those with HIV but negative Leishmania serology and were similar to those with HIV / active Leishmania coinfection.

Introduction

- The authors should mention somewhere in the introduction that the immunologic profile of people living with HIV with positive serology for L. infantum has not previously been described.

- Line 94 - IFAB should be I-FABP

- Line 94 – “consequently” should be “indicative of”

- Line 121 – “indeed” should be “furthermore” as the sentence following is about CD8+ T cells rather than CD4+ T cells in the previous sentence.

- Line 129 – The authors should specify that symptomatic VL, recovered VL and DTH+ were all people without HIV.

- Line 133 and 135 – “influenced” should be “was associated with” – the authors cannot make causative assertions based on cross-sectional data even if the literature supporting such causation is strong.

- Line 134 – suggest adding “latent or active” before “L. infantum”

- Line 135 – suggest changing “or” to “and”

- Line 137 – The authors should specify which group had a higher risk of active Leishmaniasis / HIV coinfection in comparison to which group; otherwise this sentence doesn’t have much meaning.

Methods

- Line 166 – “from which” rather than “which”

- In table 1, suggest specifying in the recovered VL and DTH+ groups that these participants were immunocompetent without HIV as it is possible to be immunocompetent with HIV (if diagnosed and commenced on treatment early).

- Line 236 – suggest adding “before any study-related procedures” after “guardian” (the authors verified this in their response to my comments but didn’t add this to the manuscript)

- Regarding exemption of consent, does this just refer to those whose only data is in the original cross-sectional study, ie who didn’t have repeat bloods? Might be worth clarifying this in the manuscript.

**Results**

-Does the analysis presented match the analysis plan?

-Are the results clearly and completely presented?

-Are the figures (Tables, Images) of sufficient quality for clarity?

Reviewer #1: (No Response)

Reviewer #2: The authors have addressed the concerns raised in my first review whenever possible, and the manuscript is strengthened by these changes.

Reviewer #3: Results

- Line 256 – please add a p value; I suspect this will be > 0.05 given that the 95% CI crosses 1.

- Line 260 – Most subjects in each group were male – incorrect – should be eg “four of the seven groups had a male predominance while the remainder were more evenly balanced”.

- Line 310 – specify that VL, recovered VL and DTH+ are without HIV (same for figure 2)

- Lines 323 to 339 – this paragraph is still very challenging to read as there are so many comparisons described many of which are not of interest. I would suggest starting with an overall comment (“groups a, b and c had the highest frequencies of x overall with no significant differences found between them”) and then focusing on the relevant/interesting findings (“all three groups had higher frequencies of x than d, e and f for both CD4+ and CD8+ T cells”). The same should be done with the subsequent paragraphs to make it easier to read and understand.

- Figure 1 – why do the p values differ from tables S4 and S5? Why have two different methods been used to do the statistical analysis? I would suggest just one method; otherwise it looks like the authors are fishing for positive results. The authors should also review the horizontal significance bars over the figures – some comparisons are clearly significant (as shown in the tables, eg UC vs AIDS/VL for CD8+CD38+HLA-DR+) but there is no horizontal bar showing this in the figure.

- Figure 2 – the same issues should be reviewed as for Figure 1 – I suggest the authors use one statistical approach for each comparison rather than two. The p values should then be fixed and the horizontal bars should be reviewed.

- Lines 326-328 – this sentence implies that this was a longitudinal analysis; I suggest the authors reword this to say x was lower than y rather than imply that something decreased after recovery.

- Lines 340-342 – I suggesting putting this the other way around, ie CD57 was higher in these groups than this group. The subsequent sentence is a repeat sentence.

- Line 358 – “senescent” should be “exhausted”

- Line 363 – The authors should include the multivariable analysis results in a table in the supplement including odds ratios and 95% CIs.

- Line 369 – The authors need to specify in the manuscript which Spearman correlations were prespecified and performed. The results of the correlations should be shown in the supplement including graphs for significant associations.

**Conclusions**

-Are the conclusions supported by the data presented?

-Are the limitations of analysis clearly described?

-Do the authors discuss how these data can be helpful to advance our understanding of the topic under study?

-Is public health relevance addressed?

Reviewer #1: (No Response)

Reviewer #2: The authors have addressed the concerns raised in my first review whenever possible, and the manuscript is strengthened by these changes.

Reviewer #3: Discussion

- Line 412 – “Thus, in HIV+ population” – the authors should say “in people living with HIV” but more importantly this sentence is a non-sequitur - the previous sentence was talking about active infection in immunocompetent individuals and this sentence is talking about latent infection in people living with HIV so the word “thus” is not justified. The authors should acknowledge in the manuscript that sensitivity of serology is unknown in this population and explain why they didn't use DTH as the screening test.

- Line 433 – “recurring” should I think be “relapsing”

- Line 440 – suggest adding a line saying that it would have been helpful to have an additional group which was positive Leishmania serology without HIV to help evaluate these questions.

- Lines 456-459 – This is a non-sequitur - what is the evidence that overall CD4+ T cell activation predicts risk of development of activated Leishmania in coinfected individuals? The only evidence that asymptomatic HIV/Leish subjects can probably control infection is that they are asymptomatic but this clearly is not necessarily a permanent state. I suggest deleting this sentence.

- Line 459-460 – “apparently influenced” should be changed to “was associated with” as no causality can be inferred by cross-sectional data.

- Line 463 – “indicates the presence of Leishmania infantum infection increase” should be changed to “suggest the presence of Leishmania infantum infection may increase” – again causality cannot be inferred.

- Line 466 – IFAB should be I-FABP

- Line 477 – “influenced” should be “was associated with”

- Line 485-486 – “CD4+ proportions” should be “CD4+ T cell counts”. I would probably leave this part of the sentence out actually as the convention is to look at the frequency of cell subsets as a proportion of the parent population rather than absolute values in blood.

- Line 496 – “we hypothesize that with progression of asymptomatic to symptomatic VL” is repeated.

- Line 497 – “could” – what could? This makes no sense – the subject is missing. If the subject is “progression of asymptomatic to symptomatic VL” then the word “with” should be removed.

- Line 504 – “immunosuppression” – no need to invoke immunosuppression here – increasing activation of the immune system would be sufficient to amplify disease progression in the absence of ART.

- Lines 510-511 – the authors should specify what else they could have used in addition to serology (eg ?DTH). Using PCR to detect latent disease makes no sense – I would presume that a positive PCR would indicate active infection.

- Lines 514-515 – This is a non-sequitur - why do the authors decide to call PLHIV with positive serology asymptomatic based on symptomatic pts being less likely to have positive serology? They are asymptomatic based on being asymptomatic. The authors should reword this to make it logical or delete it.

- An additional line should be inserted in the limitations paragraph saying that another limitation was that people with positive Leishmania serology but without HIV were not included and could have helped to evaluate whether HIV and Leishmania were having synergistic effects.

- Line 519 – the authors should expand on the difference between groups in ART adherence and how this may entirely account for the difference between HIV and asymptomatic HIV/Leish groups in activation / exhaustion / senescence findings.

- Line 520 – “can lead to” should be “is associated with” – the authors can’t infer causality as stated in the previous paragraph!

- Line 522 – “temporary” – what are the authors basing this assertion on? This is a cross-sectional study. If there is longitudinal data supporting this statement then it should be referenced here.

- Line 523 – “after treatment” – after what treatment? For HIV or for Leishmania?

- Lines 529-532 – I know it was me who suggested this line but it is actually quite a weak way to end the article as the implication is that we should be focusing our efforts on working out which patients to treat for latent Leishmaniasis when in fact these patients should all be on ART for its mortality benefit based on international guidelines. I would suggest changing to “Longitudinal trials are needed to determine whether, in the setting of universal ART and sustained viral load suppression, T cell activation and exhaustion are seen in people living with HIV with positive serology for Leishmania and, if so, whether these immune parameters predict active VL despite suppressed virus”. This is, after all, what the authors are now doing.

**Editorial and Data Presentation Modifications?**

Reviewer #1: (No Response)

Reviewer #2: None

Reviewer #3: (No Response)

**Summary and General Comments**

Reviewer #1: The authors did not satisfactorily address this reviewer's comments, particularly contextualizing their findings with the vast literature on exhaustion and senescence. It does not look like the authors understand the complex literature on exhaustion or failed to demonstrate it in their manuscript. For example, PD-1 is induced by TCR engagement, and, therefore, activated T cells will express PD-1 merely by seeing antigen. Long-term exposure to antigen will generate exhausted T cells. However, there is an array of inhibitory receptors, and PD-1 is just one of them. What creates a major issue is not using PD-1 as a marker of exhaustion, but it is that the authors failed to contextualize and address the caveats to their findings. As exhaustion, and senescence are central components of the manuscript and part of the title of the paper, altering the manuscript was absolutely necessary and the authors failed to thoughtfully incorporate the criticism.

Reviewer #2: The authors have addressed the concerns raised in my first review whenever possible, and the manuscript is strengthened by these changes.

Reviewer #3: The suggestions are outlined above. There is still a lot of rewording required and the authors should include the results of the multivariable and correlation analyses in the supplement.

PLOS authors have the option to publish the peer review history of their article (what does this mean? ). If published, this will include your full peer review and any attached files.

**Do you want your identity to be public for this peer review?** For information about this choice, including consent withdrawal, please see our Privacy Policy .

Reviewer #1: No

Reviewer #2: No

Reviewer #3: No
---

## [Editor Report · Decision Letter 2]

15 Dec 2024

PNTD-D-24-00260R2

T-cell activation, senescence, and exhaustion in asymptomatic HIV/Leishmania infantum co-infection

Dear Dr. Jeronimo,

Thank you for submitting your manuscript to PLOS Neglected Tropical Diseases. After careful consideration, we feel that it has merit but does not fully meet PLOS Neglected Tropical Diseases's publication criteria as it currently stands. Therefore, we invite you to submit a revised version of the manuscript that addresses the points raised during the review process.

Please submit your revised manuscript within 60 days. If you will need more time than this to complete your revisions, please reply to this message or contact the journal office at plosntds@plos.org. Please include the following items when submitting your revised manuscript:

We look forward to receiving your revised manuscript.

Kind regards,

Johan Van Weyenbergh

Academic Editor

Susan Madison-Antenucci

Section Editor

Shaden Kamhawi

co-Editor-in-Chief

Paul Brindley

co-Editor-in-Chief

**Additional Editor Comments :**

The editors and reviewers have extensively discussed the manuscript in its current an previous versions, hence the delay in formulating a decision. Two major points remain crucial: overstressing the exhaustion message and adding a PCA analysis that does not answer the scientific hypothesis.

1. Reviewer #1 rightfully critiqued the overselling of the “exhaustion” message based on a single marker PD-1, which is also a marker or early T-cell activation, and specifically asked to discuss this extensively. The authors only reply with a short paragraph, which is poorly formulated. They also do not mention using a single marker (PD-1) as a major limitation, as requested, and did not remove the claim of T-cell exhaustion from the title, abstract or any other part of the manuscript. Considering PD-1 is also a marker of T-cell activation, this is also in keeping with the authors’ working hypothesis in the abstract: “We hypothesized that, relative to those with HIV alone, people with co-infection would have altered T cell activation which could impact on the risk of VL.”

2. The authors claim that PCA analysis discriminates between Leishmania-infected and non-infected individuals. First, this is unsurprising since anti-Leishmania serology was one of the few parameters included to define PCs. Second, the authors did not perform a cluster analysis and/or test cluster quality (silhouette score or similar), due to the spreading of patients it seems unlikely that clusters can be discriminated. Third, the authors only analyzed PC1 and PC2, although they do not explain 75% of the variance, and they do not show the possible differential loadings of humoral and cellular parameters on PC1 and PC2 (or PC3-PC4).

Since the total review process has been extremely long, and the manuscript has scientific merit in the editors’ opinion, we would like to give the authors the chance to address these two points thoroughly in a (final) revision.

**Journal Requirements:**

At this stage, the following Authors/Authors require contributions: Carolina de Oliveira Mendes Aguiar, and Selma M.B. Jeronimo. Please ensure that the full contributions of each author are acknowledged in the "Add/Edit/Remove Authors" section of our submission form.

2) Tables should not be uploaded as individual files. Please remove these files and include the Tables in your manuscript file as editable, cell-based objects. For more information about how to format tables, see our guidelines:

https://journals.plos.org/plosntds/s/tables 

3) We have noticed that you have a list of Supporting Information legends in your manuscript. However, there are no corresponding files uploaded to the submission. Please upload the supplementary tables as separate files with the item type 'Supporting Information'.

4) We notice that your supplementary figure is uploaded with the file type 'Figure'. Please amend the file type to 'Supporting Information'. Please ensure that each Supporting Information file has a legend listed in the manuscript after the references list.

5) Thank you for stating that "All data are presented and the results will be available." We strongly recommend all authors decide on a data sharing plan before acceptance, as the process can be lengthy and hold up publication timelines. Please note that, though access restrictions are acceptable now, your entire data will need to be made freely accessible if your manuscript is accepted for publication. This policy applies to all data except where public deposition would breach compliance with the protocol approved by your research ethics board. If you are unable to adhere to our open data policy, please kindly revise your statement to explain your reasoning and we will seek the editor's input on an exemption. Please be assured that, once you have provided your new statement, the assessment of your exemption will not hold up the peer review process.

6) Please amend your detailed Financial Disclosure statement in the online submission form. This is published with the article. It must therefore be completed in full sentences and contain the exact wording you wish to be published.

7) Your current Financial Disclosure states, "Brazilian Research Council - CNPQ - universal 406076/2021-9 and National Institute of Science and Technology of Tropical Diseases (INCT-DT, 465229/2014-0)".

However, your funding information on the submission form indicates only one funder. Please ensure that the funders and grant numbers match between the Financial Disclosure field and the Funding Information tab in your submission form. Note that the funders must be provided in the same order in both places as well.                                             . 

Please indicate by return email the full and correct funding information for your study and confirm the order in which funding contributions should appear. Please be sure to indicate whether the funders played any role in the study design, data collection and analysis, decision to publish, or preparation of the manuscript.

**Reviewers' Comments:**

**Figure resubmission:**
---

## [Editor Report · Decision Letter 3]

16 Jan 2025

Dear Dr. Jeronimo,

We are pleased to inform you that your manuscript 'T-cell activation and senescence in asymptomatic HIV/Leishmania infantum co-infection' has been provisionally accepted for publication in PLOS Neglected Tropical Diseases.

Best regards,

Johan Van Weyenbergh

Academic Editor

Susan Madison-Antenucci

Section Editor

Shaden Kamhawi

co-Editor-in-Chief

Paul Brindley

co-Editor-in-Chief

---

## [Editor Report · Acceptance letter]

Dear Prof. Jerônimo,

We are delighted to inform you that your manuscript, "T-cell activation and senescence in asymptomatic HIV/Leishmania infantum co-infection," has been formally accepted for publication in PLOS Neglected Tropical Diseases.

Best regards,

Shaden Kamhawi

co-Editor-in-Chief

Paul Brindley

co-Editor-in-Chief
